# Chronic Intestinal Disorders in Humans and Pets: Current Management and the Potential of Nutraceutical Antioxidants as Alternatives

**DOI:** 10.3390/ani12070812

**Published:** 2022-03-23

**Authors:** Giorgia Meineri, Elisa Martello, Elisabetta Radice, Natascia Bruni, Vittorio Saettone, David Atuahene, Angelo Armandi, Giulia Testa, Davide Giuseppe Ribaldone

**Affiliations:** 1Department of Veterinary Sciences, School of Agriculture and Veterinary Medicine, University of Turin, 10095 Grugliasco, TO, Italy; giorgia.meineri@unito.it (G.M.); vittorio.saettone@unito.it (V.S.); david.atuahene@unito.it (D.A.); 2Division of Epidemiology and Public Health, School of Medicine, University of Nottingham, Nottingham NG5 1PB, UK; 3Department of Surgical Sciences, Medical School, University of Turin, 10124 Turin, TO, Italy; elisabetta.radice@unito.it; 4Candioli Pharma S.r.l., 10092 Beinasco, TO, Italy; natascia.bruni@candioli.it; 5Department of Medical Sciences, University of Turin, 10124 Turin, TO, Italy; angelo.armandi@unito.it (A.A.); gi.testa@unito.it (G.T.); davidegiuseppe.ribaldone@unito.it (D.G.R.)

**Keywords:** inflammatory bowel disease, ulcerative colitis, chronic enteropathies, phytocomplex, trace elements, vitamins, nutraceuticals

## Abstract

**Simple Summary:**

Chronic disorders of the intestinal tract (CID) are characterized by signs of inflammation of the intestine for a period of at least three weeks. Both humans and pets can be affected by these disorders. Different therapeutic approaches can be selected to treat patients and the use of natural products has been increased in the last decade, since oxidative stress plays a key role in the progression of the chronic intestinal disorders. In this review, the antioxidant proprieties of several natural products with potential for treatment of CID in human and veterinary medicine are highlighted. Unfortunately, few clinical trials report the use of these products for treating CID in humans and none in animals.

**Abstract:**

Chronic intestinal disorders (CID) are characterized by persistent, or recurrent gastrointestinal (GI) signs present for at least three weeks. In human medicine, inflammatory bowel disease (IBD) is a group of chronic GI diseases and includes Crohn’s disease (CD) and ulcerative colitis (UC). On the other hand, the general term chronic enteropathies (CE) is preferred in veterinary medicine. Different therapeutic approaches to these diseases are used in both humans and pets. This review is focused on the use of traditional therapies and nutraceuticals with specific antioxidant properties, for the treatment of CID in humans and animal patients. There is strong evidence of the antioxidant properties of the nutraceuticals included in this review, but few studies report their use for treating CID in humans and none in animals. Despite this fact, the majority of the nutraceuticals described in the present article could be considered as promising alternatives for the regular treatment of CID in human and veterinary medicine.

## 1. Introduction

Chronic intestinal disorders (CID) are a common cause of persistent or recurrent gastrointestinal (GI) signs extended for more than three weeks.

Inflammatory bowel disease (IBD) in humans is a group of chronic diseases of the gastrointestinal (GI) tract. They include Crohn’s disease (CD) and ulcerative colitis (UC). The peak of onset is at the age of around 20–40 years old, but they can occur at all ages, last for a lifetime, and severely affect the quality of life [1]. CD can involve the entire GI tract from the mouth to the anus and is characterized by deep lesions of the GI wall. Symptoms include diarrhea, abdominal pain, and weight loss. CD is burdened by complications like stenosis, abscess, and perianal involvement [2]. On the other hand, UC typically affects the rectum and can involve the colon in a continuous way. It is characterized by lesions of the mucosa (erythema, erosions, and ulcers) and clinically followed by bloody diarrhea [3]. Unfortunately, the etiological factors triggering IBD are not yet fully elucidated, but genetic predisposition, gut microbiota dysbiosis, dysregulated immune response, and environmental factors (including diet) are thought to be involved in the pathophysiology mechanisms [4].

In veterinary medicine, the use of the term chronic enteropathies (CE) is preferred instead of IBD to identify a group of idiopathic intestinal disorders with evident GI signs (recurrent or chronic), and inflammation in the lamina propria of the small intestine, large intestine, or both [5]. Nonetheless, several phenotypes of IBD have been identified in dogs [6]. In fact, IBD in dogs has different forms when compared to humans, where more standardized clinical, endoscopic, and pathologic aspects can be found [7,8]. CE can be classified retrospectively based on the response to treatment into (1) food-responsive enteropathy, (2) antibiotic-responsive enteropathy, (3) immunosuppressant-responsive enteropathy, and (4) protein-losing enteropathy [9]. Most forms of CE involve a complex interplay among host genetics, the intestinal microenvironment (including bacteria and dietary patterns), and the immune system [10]. The clinical signs often overlap, but they can be distinguished into large intestinal localization (dyschezia, tenesmus, increased frequency of defecation, small volume of faeces, mucus, and blood), small intestinal localization (large volume diarrhoea, weight loss, and vomiting), melaena (upper GI bleeding/ulceration), and abdominal pain, which is uncommon in chronic enteropathy and raises suspicion of pancreatic disease, structural disorders, or perforation. A breed predisposition for CE in dogs has also been described [11]. Different CE phenotypes may reflect different disease severity affecting the intestinal immune system and varied response to treatment over time. This makes it difficult to standardize a treatment plan for these animals. With respect to CE treatment for dogs and cats, there is strong evidence that controlled, elimination, and hydrolyzed diets are beneficial [12]. In addition, the traditional choice of drugs in dogs and cats with CE includes anti-inflammatories, antibiotics, immunosuppressive, and other medications [9]. 

The use of complementary and alternative medicine is gaining increasing evidence in both humans and pets with digestive disorders. For instance, some commonly used treatments include probiotics, prebiotics, omega-3-fatty acids, vitamins, minerals, bioactive peptides, colostrum, aloe vera, and turmeric. These novel approaches may improve human and animal medical conditions in case of chronic intestinal disorders [9,12].

This review will be focused on both traditional therapies and nutraceuticals with specific antioxidant properties that possess established or promising effectiveness for the treatment of CID in humans and animals. 

## 2. Oxidative Stress-Induced Damage in Chronic Intestinal Disorders

Cell inflammation and oxidative reactions caused by activated leukocytes producing excessive reactive oxygen species (ROS) can overpower the tissue’s antioxidant defenses, resulting in a dysfunction of the enteric mucosa. As part of basal metabolic function, ROS are produced by numerous enzymatic reactions in various cell compartments, including the cytoplasm, endoplasmic reticulum, cell membrane, peroxisome, and mitochondria. ROS plays a variety of physiological roles, including the control of cell differentiation and development, apoptosis, and inflammatory processes via cell signaling [13]. A complex, dynamic mechanism, where different molecules undergo well-established oxidation–reduction reactions, maintains the homeostasis of the intestinal ecosystem, which represents the intestinal mucosa’s response to prevent oxidative damage. The ROS generated by unstable types of oxygen—superoxide anion, hydrogen peroxide (H_2_O_2_), and hydroxyl radicals—are the key pro-oxidant molecules. The pathogenesis of CE has been linked to oxidative stress, which may be a key effector mechanism leading to molecular/cellular damage and tissue injury. ROS promote cell damage by preventing the accumulation of antioxidant defenses in cells. For example, oxidative damage is observed in CD patients’ intestinal mucosa as well as their peripheral blood leukocytes [14]. Immune cells that enter the mucosa produce a number of ROS that can be harmful to tissue integrity. Patients with CD have lower levels of antioxidant vitamins A, C, E, and beta-carotene in their blood and mucosa, as well as lower activity of the major cellular antioxidant enzymes glutathione peroxidase (GPx) and superoxide dismutase (SOD) [15]. Oxidative stress and redox signaling are intimately involved in the upregulation of inflammatory cytokines as well as in the increased infiltration of inflammatory cells, through the stimulation of signaling pathways (especially the redox-sensitive transcription factor, nuclear factor kappa-light-chain-enhancer of activated B cells). Inflammation also enhances oxidative stress by inducing the development of ROS and the release of myeloperoxidase from inflammatory cells [16]. In the literature, there is evidence indicating the role of oxidative stress in humans with IBD and recent studies suggest that it could also be a relevant factor in the pathogenesis of dogs affected by IBD [17].

## 3. Therapeutic Management of Chronic Intestinal Disorders

Current therapies for IBD in humans are based on shared guidelines, particularly those published in the United States and Europe [18,19,20]. Patients affected by IBD are normally treated with a step-up approach, starting with mesalazine and gradually adding higher-level drugs until a complete clinical, laboratory, and endoscopic remission is achieved. The step-up approach was the one classically used also for CD, but in recent years the top-down paradigm has been established, starting the most effective drugs available in patients with predictive factors of more aggressive disease (Figure 1).

In veterinary medicine, new approaches to the management of CID in dogs and cats have been developed over the last 30 years. However, therapy for CE is difficult to establish, since the pathogenesis of the disease is not easily understood. The choice depends on the seriousness of the disease and the response to drugs. The most used therapies are antimicrobials, immunosuppressants, and anti-inflammatory drugs. Unfortunately, the scientific evidence of the efficacy and effectiveness of these drugs in animals is lacking and variable across studies. There is stronger evidence for the use of controlled, elimination, and hydrolyzed diets, which are the first-choice approach for pets [5] (Section 4, Figure 1). When deciding which treatment is the most appropriate, adding symptomatic measures (gastroprotectors, antiemetics, motility modulators, etc.) would be beneficial to correct any imbalance.

### 3.1. Humans

#### 3.1.1. Conventional Therapy

Mesalamine represents the cornerstone of therapy of UC. Mesalamine is able to induce and maintain remission. It is most prescribed for UC with mild or moderate disease symptoms. Topical (rectal) plus oral mesalamine is the most efficacious mode of administration. Mesalamine’s exact function is unclear, however the most accepted theory is that it reduces the synthesis of prostaglandins and leukotrienes by modulating the inflammatory response. Mesalamine is also thought to be an antioxidant and a free radical scavenger [21].

Systemic corticosteroids (particularly prednisone and methylprednisolone) are the therapy of choice in patients with severe disease activity or with mesalamine failure in UC. In case of mild or moderate disease activity, low-absorption corticosteroids are available: budesonide for CD, beclomethasone, or budesonide multi matrix (MMX) for UC [22]. Steroids have a good rate of effectiveness (around 80%), but unfortunately they cannot be used for prolonged times (no more than 3–4 months) as they are characterized by numerous side effects and lose effectiveness over time [23].

After initial steroid-induced remission, immunosuppressants are used to achieve long-term steroid-free remission [24]. The immunosuppressants used in IBD include thiopurine (azathioprine, 6-mercaptopurine), methotrexate (mainly in CD), and cyclosporine (mainly in UC). 

Another approach to improve gut health and to treat IBD includes the use of probiotics, and prebiotics [25].

#### 3.1.2. Advanced Therapy

Biological drug therapy (target therapy) has revolutionized therapy for IBD and other autoimmune and neoplastic diseases over the last 25 years.

Anti-tumor necrosis factor (TNF) therapy has been widely used for the treatment of IBD in the last two decades as a new approach to the disease’s management. Anti-TNF therapies approved for the treatment of both CD and UC, include infliximab and adalimumab. Golimumab has also been evaluated for use in UC [26]. Unfortunately, one-third of patients are primary non-responders, and dose intensification is needed in 23 to 46% of responders, with drug discontinuation occurring in 5% to 12% of patients per year [27].

Vedolizumab is a humanized monoclonal antibody and is the first biological drug created to be selectively effective on the bowel [28].

Ustekinumab is a human monoclonal antibody and it is the latest biological drug introduced for the treatment of IBD, borrowed from the excellent experience in psoriasis [29].

Tofacitinib (anti-JAK) is the latest drug available to treat IBD, in particular UC. Blocking the JAK-STAT pathway, it interferes with the signaling pathways of several cytokines [30]. 

### 3.2. Dogs and Cats

#### Conventional Therapy

Antimicrobials are used to fight the microbial dysbiosis which may initiate and drive host inflammatory responses in animals with CE. This therapy is often associated with diet and other drugs, leading to difficulties in interpreting the effectiveness of a single antimicrobial product. For example, studies have supported the efficacy of rifaximin and oxytetracycline or of metronidazole and tylosin with an additional anti-inflammatory action [9,31,32,33]. If an antimicrobial treatment is not successful within two weeks, a new therapy should be considered, with the aim of achieving long-term control of the disease. This would also avoid the risk of developing antimicrobial resistance. In addition, interest in new antimicrobial alternatives to manipulate the GI microbiome is growing [5].

A small percentage of animals affected by CE need administration of immunosuppressants. Following the failure of diet and antibiotic treatments, the administration of immunosuppressants could be successful. When a good response to this treatment exists the pathological condition is defined as immunosuppressant-responsive CE. On the other hand, dogs not responding to this treatment are categorized as having non-responsive enteropathy. A treatment plan with immunosuppressive drugs usually includes the use of these medications and modification of the diet [34]. Commonly used immunosuppressant drugs are azathioprine and cyclosporine [33,35].

The use of anti-inflammatory drugs helps control the inflammation of the intestine in animals with CE. They usually work together with other therapeutic approaches such as diet and antimicrobial agents. The most commonly used anti-inflammatory drugs are glucocorticoids, 5-aminosalicylates [33]. As reported before, there are a few antimicrobials with anti-inflammatory properties like metronidazole and tylosin [33].

In addition, the use of prebiotics and probiotics improves the gut microbiota of animals, especially when affected by GI diseases [25].

## 4. Dietary Interventions in Chronic Intestinal Disorders

In the medical community, there is currently no consensus on dietary recommendations for adult patients with IBD. It is difficult to make strong recommendations due to the lack of randomized controlled trials investigating specific diets and eating habits. Exclusive enteral nutrition (EEN) is an exception since it is prescribed as a first-line treatment for children and adolescents with acute active CD to promote remission [36]. The European Society for Clinical Nutrition and Metabolism (ESPEN) advises that during remission periods, no particular diet should be undertaken because it does not seem to be successful in maintaining remission [37]. Several dietary compounds have been identified as influencing the development and maintenance of IBD, while others appear to be protective.

In veterinary medicine, the dietary approach is the first choice to control the symptoms of CE in pets [12]. Several studies demonstrated the efficacy of diet manipulation that in many cases results in a promising long-term outcome (>6 months). In some more severe cases, where a long-term positive effect is not observed, antibiotic or immunomodulant treatments should be added [5]. The change in diet alone has been reported to be effective in over 50% of CE cases as reported in a previous review [38]. Despite the fact that most of the study protocols have low-quality designs, long-term response seems to be supported by diet when used as first-line treatment. The GI tract can be damaged when a subject is affected by CE (microbiota, intestinal permeability and motility, and mucosal immune system). As a consequence, choices in nutrition can influence the equilibrium of GI components and functions. However, well-designed studies are still needed to determine whether specific dietary elements could represent risk factors for the development of CE in dogs and cats. A recent review by Kathrani [12] exhaustively reported the use of diet to manage CE in dogs and cats.

### 4.1. Humans

#### 4.1.1. Fiber Content

Howeler created the term “dietary fiber” to characterize a complicated group of non-digestible components of cell walls [39]. Non-starch polysaccharides (e.g., pectin, cellulose), non-carbohydrate-based polymers (e.g., lignan), resistant oligosaccharides (e.g., galatooligosaccharides, fructooligosaccharides), and animal-derived carbohydrates (e.g., chitin) have all been lumped together under the term [40]. Unlike most dietary components, non-digestible dietary carbohydrates (resistant starch and fiber) can withstand stomach acidity and do not degrade in the human small intestine; instead, they are fermented by the gut microbiota consortium within the large intestine, where one microbe initiates the fermentation process and others continue it, resulting in a systematic process [41].

Dietary fiber is made up of a variety of connected monosaccharides that form a variety of molecules with different side chains and physical properties, such as solubility and physical organization. While dietary fibers can be classified in a variety of ways, the most prevalent method for nutritional purposes in humans divides them into water-soluble and insoluble fibers [42]. The degree of fermentation by gut bacteria is related to water solubility in the gastrointestinal system. Soluble dietary fiber can reduce glycemic response by increasing digesta viscosity, which delays stomach emptying and nutrient release. Easily digestible fibers (Arabinoxylan, pectin, inulin, b-glucans, fructo-oligosaccharides, xyloglucans, and galactooligosaccharides) are examples of soluble dietary fibers and are degraded to volatile fatty acids, which serve as a nutrient substrate for the microbiota as well as the enterocytes [43]. Insoluble dietary fibers, such as lignin and cellulose, are thought to be less beneficial to gut microorganisms because their strong hydrogenbinding networks restrict the amount of surface area available for fermentation. Fibers that are difficult to digest stimulate intestinal peristalsis and thus the expulsion of pathogenic microorganisms [44].

Although the literature on the relationship between dietary components and the onset of IBD is still uncertain, several studies showed that dietary fiber intake has a positive impact and plays an important role in the prevention of CD [45]. Anti-inflammatory action through butyrate’s protective effects, reduction in colonic permeability, and prevention of pro-inflammatory cytokine transcription are some of the mechanisms proposed in the literature [46]. Furthermore, dietary fibers have shown an effect on the microbiome, influencing immunological homeostasis in a regulatory manner [47]. Fiber has a prebiotic function by increasing the growth of beneficial bacteria [12]. Patients with IBD, on the other hand, often complain that high-fiber foods aggravate their symptoms [48] and fiber is contraindicated in patients with stenosing CD.

#### 4.1.2. Specific Diets

In the existing scientific literature, complex diets for IBD treatment have been suggested.

##### Exclusive Enteral Nutrition (EEN) 

EEN relies on the administration of a liquid nutrient formula orally or via a feeding tube for 4–12 weeks. Individual amino acids are found in elemental formulas; semi-elemental peptides of different chain lengths are found in semi-elemental formulas; and intact proteins are found in polymeric formulas. Food reintroduction data is still sparse and inconclusive after this time frame. Most centers, on the other hand, suggest a 2–3 week gradual reintroduction of the normal diet [49]. This type of diet is useful in children with CD, but it is not useful in UC [50]. However, since this dietary therapy requires children to abstain from eating for many weeks, adherence is challenging and unpredictable. This type of diet can have a big impact on the child’s family as well. In addition, essential amino acids have been shown to activate mucosal immunity while maintaining intestinal homeostasis and forming the intestinal mucosal barrier [51]. During EEN therapy, there could be a decrease in the development of metabolites that may be involved in the immunological attack on gut microbiota [52]. 

##### Carbohydrate Diet (SCD)

SCD is a diet for patients affected by IBD. This diet consists of a modified carbohydrate diet that allows monosaccharides but prohibits disaccharides and most polysaccharides. Fruits and vegetables with more amylose than amylopectin, dry-curd cottage cheese, butter, nuts, nut-derived flours, meats, eggs, and oils are allowed in the SCD. Sucrose, isomaltose, maltose, lactose, both real and pseudo-grains and grain-derived flours, okra, potatoes, fluid milk, corn, soy, lactose-rich cheeses, and most food additives are not permitted. To avoid lactose, the diet is supplemented with entirely fermented yogurt [53]. SCD is focused on the ingestion of complex carbohydrates with limited digestive requirements. Although the mechanism of action is unknown, it is hypothesized that changes in the fecal microbiome reduce intestinal inflammation. A very restrictive diet necessitates significant lifestyle changes, and patients need follow-up to ensure proper nutrition, taking into account that specific dietary deficiencies can occur as a result of specific food restrictions, especially of dairy products that contain vitamin D and calcium. The limited intake of grains, fruits, and vegetables can also result in folate, thiamine, and vitamins B6, C, and D deficiencies.

##### Anti-Inflammatory Diet for IBD (IBD-AID) 

IBD-AID was developed by a team at the University of Massachusetts Medical School and is based on the SCD. This diet was created for patients who had failed to respond to pharmacological treatment [54]. The IBD-AID is made up of five parts: the first is the modification of carbohydrates, such as refined or processed complex carbohydrates and lactose; the second is the ingestion of probiotics and prebiotics; the third is the modification of dietary fatty acids; the fourth is the detection of the overall dietary pattern and missing nutrients, as well as the identification of intolerances; and the fifth is the modification of dietary fat acids. Likewise, this is a very restrictive diet, that can lead to nutritional deficiencies due to the lack of nutrients, especially micronutrients. Its mechanism of action has yet to be discovered.

### 4.2. Dogs and Cats

#### 4.2.1. Fiber Content

The dietary use of high fibers has been found to have numerous health benefits including anti-inflammatory properties and helping to maintain the intestinal barrier function. Fiber has a prebiotic function in promoting the growth of beneficial microorganisms in the intestine [12]. 

#### 4.2.2. Specific Diets

In the scientific literature, different diets as treatments for CE in dogs and cats have been suggested.

##### Hydrolyzed Diets

Hydrolyzed diets have been successfully used in the management of CE in dogs and cats. In this type of diet, proteins are broken down by enzymes and the organisms do not recognize them as proteins. This type of diet can help to reduce the level of dysregulation of the immune system and is considered highly digestible [55].

##### Limited-Ingredient Diets

A limited-ingredient diet should ideally provide a single carbohydrate and a single protein source. Considering the exposure of dogs and cats to multiple ingredients in their diets, a limitation of the number of ingredients could be beneficial to limiting the antigen load in the GE tract in order to reduce intolerances [55].

##### Fat Reduced Diets

Reduction of fat in the regular diet reduces the passage of the fat in the colon, reducing dysbiosis and epithelial cell damage. A fat restriction diet has been shown to be effective in dogs affected by PLE-lymphangiectasia but it could also be considered as an option in cases of CE [55]. The fat content of the dry matter must be no more than 15% in dogs and 25% in cats [56].

##### Gluten-Free Diets

A gluten-free diet seems to be quite effective in reducing GE symptoms even though no specific trials have been performed yet. The effectiveness of this diet has been hypothesized because some of the commercial diets are already gluten-free. A novel protein diet has been found to be more effective in pathologies involving the large intestine [49]. 

##### Parenteral Nutrition (PN)

This type of approach is an option that is required only in rare cases, being very expensive and of difficult management for the pet owners. This type of nutrition keeps under control the daily intake of the different nutrients and contributes to bowel rest. This technique is used in human medicine even if the enteral nutrition results in a more physiological intervention for the intestine [12].

## 5. Role of Nutraceuticals as Antioxidants in Chronic Intestinal Disorders: Phytocomplex, Trace Elements, and Vitamins 

Functional foods and bioactive natural compounds have become relevant research topics when discussing new approaches to treat intestinal disorders in human and veterinary medicine, as the long-term use of traditional drugs causes complications. 

Nutraceuticals by definition, are food or food supplements that have been formulated or processed to enhance the pharmacological properties of functional bioactive ingredients, providing health and medical benefits, including the prevention and treatment of diseases [57]. 

Phytochemicals of nutraceutical importance, are non-nutritive plant chemicals found in fruits and vegetables [50,58,59]. Plants produce phytochemicals as part of their defense mechanisms against pathogens, which are derived from their primary and secondary metabolisms. These chemicals play a significant role in the body’s defense against oxidative stress and inflammation together with other potential health benefits.

Complementary and alternative medicine (CAM) use has been found to be more common in human patients with IBD than in healthy individuals, with some studies reporting values as high as 72% [60]. CAM can assist in the treatment of chronic intestinal disorders and prolong the clinical remission in human patients because of the antioxidant and anti-inflammatory properties of the used plants. Several studies in dogs and cats reported the use of these natural ingredients as promising to manage diseases. 

Nutritional supplements described in this section include phytocomplex, vitamins, and minerals. Here, available evidence of their proven or promising effectiveness in human and veterinary medicine is presented (Table 1).

### 5.1. Phytocomplex

#### 5.1.1. *Curcuma longa*

*Curcuma longa* is a perennial herb which, when dried, becomes the source of the spice turmeric. Turmeric’s active part is the flavonoid curcumin. Water and fat-soluble turmeric and curcumin show strong antioxidant properties. Since curcumin has autophagy-regulating properties it helps in the improvement of colitis. Curcumin also inhibits the development of autophagosomes in colonic epithelial cells and has also an anti-inflammatory effect in acute and chronic inflammation status [61].

The administration of curcumin was found to be more effective than placebo in keeping human patients with quiescent UC in remission [62] and with various pro-inflammatory diseases [63]. A dosage range of 1500–3000 mg/day demonstrated a significant difference between clinical remission and endoscopic remission rates in the intervention and placebo groups [62]. Only Hanai et al. reported mild side effects in a Japanese population, such as abdominal distension, nausea, and an increased number of bowel movements [64].

The veterinary use of curcumin has also shown promise in intestinal diseases in dogs with stimulation of the antioxidant system and evidence of anti-inflammatory effects [65,66]. The findings of a study suggested that curcumin and the commercial product Meriva curcumin phytosome^®^ reduced inflammation in canine IBD but no specific antioxidant effect was tested [67]. No data on cats is available.

#### 5.1.2. *Aloe vera*

*Aloe vera* is a tropical, drought-resistant succulent plant. The leaves are filled with brown or yellowish milky juice that contains the most bioactive compounds. Not all species are therapeutic, others can be toxic or neutral. It has been shown to have antioxidant, antibacterial, anti-inflammatory, immune-boosting, anti-cancer, healing, and anti-diabetic effects. The 75 biologically active compounds (i.e., flavonoids, terpenoids, lectins, magnesium, zinc, vitamins) present in *Aloe vera* show synergic effects [68,69]. 

A randomized, double-blind, placebo-controlled trial of oral *Aloe vera* in human patients affected by UC has been performed [70]. The volume was 100 mL twice daily, which is the greatest quantity that may be tolerated and is widely used by people who use aloe vera gel for a variety of purposes. To ensure tolerance and reduce the risk of adverse effects, patients were advised to start with 25–50 mL twice daily for up to 3 days. Oral *Aloe vera* was found to produce a clinical response more frequently than placebo, as well as a reduction in histological disease activity. Among the 30 patients randomized to *Aloe vera* gel, no serious adverse events were registered; however, nine patients reported of abdominal bloating, one of foot pain, one of sore throat, one of temporary ankle swelling, one of acne, and one of worsening eczema.

One study reported that *Aloe vera* juice was used as a stomach tonic for vomiting and irritation in dogs [71]. To our knowledge, no specific studies on the use of this nutraceutical in the treatment of CE have been performed in veterinary medicine.

#### 5.1.3. *Boswellia serrata*

The gum resin obtained from the *Boswellia serrata* (*B. serrata*) tree, a species of the Burseraceae family, is known as frankincense. The major active derivatives are 11-keto-β-boswellic acid, boswellic acid, and acetyl-11-keto-β-boswellic acid, all of which are believed to have antioxidant and anti-inflammatory properties [72]. In India, preparations made from the gum resin of *B. serrata* have been used as a common remedy for inflammatory diseases in Ayurvedic medicine. 

For example, *B. serrata* extract was successfully used in maintaining the remission phase over a prolonged period of time in human patients with UC [73]: Casperome^®^ is a delivery form containing a 1:1 ratio of highly standardized *B. serrata* extract and soy lecithin, as well as around half a part of microcrystalline cellulose to improve the physical condition and standardize the product to a content of triterpenoid acids of at least 25%. Remission or improvement in one or more of the parameters was achieved by administering *Boswellia* gum resin [74]. This nutraceutical is largely demonstrated to be effective as an anti-oxidant, anti-inflammatory, and anti-diabetic agent in dogs in both in vitro and in vivo studies [75,76,77]. No studies on the use of *B. serrata* in the management of CE in dogs and cats have been performed so far.

#### 5.1.4. *Triticum aestivum*

Wheatgrass juice (*Triticum aestivum*) is an extract made from wheatgrass pulp. Wheatgrass extracts showed antioxidant activity by scavenging free radicals in conjunction with phenolic and flavonoid material. 

In a study, for one month, 100 mL wheat grass juice was consumed. The juice was to be consumed right away by the patients. The doses were gradually raised, starting with a 20 mL initial dose and increasing by 20 mL every day. It was proven to be an efficient and safe treatment for active distal UC as a single or adjuvant treatment [78]. This natural product has been demonstrated to be an effective antioxidant in reducing senile cataracts in dogs [79]. The biologically active substances can be partially absorbed during digestion and future use in treating GI conditions in animals can be considered. 

#### 5.1.5. *Plantago* spp.

*Plantago ovata* is an annual herb, local to the Mediterranean region. It is a source of soluble fiber and has been reported to treat intestinal disorders like diarrhea, constipation, IBD, and hemorrhoids [80]. Polysaccharides, flavonoids, phenolic compounds (caffeic acid derivatives), terpenoids, alkaloids, and vitamins are among the bioactive components [81].

Its seeds (10 g b.i.d.) have been shown to be as effective as mesalazine in preventing UC relapse. These treatment choices may be appealing to human patients who are unable to tolerate mesalazine [82]. 

*Plantago major* seems to be effective in the complementary management of UC [81]. 

*Plantago* spp. can be used to treat endoparasites and stomach problems in animals [71,83], but no specific studies on CE in animals have been carried out.

#### 5.1.6. *Serpylli herba*

*Serpylli herba* is a European Pharmacopeia officinal medicine made up of the aerial portions of wild thyme (*Thymus serpyllum*). It has been tested in rodent colitis experimental models [84]. In two separate experimental models of colitis in rats (TNBS) and mice (DSS), *S. herba* extract showed intestinal anti-inflammatory properties, indicating that it could be used to treat human IBD [84]. The antioxidant qualities are most likely attributable to the high polyphenol content. Given the in vitro results, its efficacy on intestinal disorders in human and veterinary medicine cannot be excluded even though no in vivo trials have been performed to date.

#### 5.1.7. *Vaccinium myrtillus*

Bilberries (*Vaccinium myrtillus*) contain one of the highest levels of natural anthocyanins that exert the most effective antioxidant activity [85]. It inhibits protein and lipid oxidation. Human patients with mild to moderate UC were treated with an anthocyanin-rich bilberry preparation in addition to their regular medicine in an open-label pilot study [86]. The bilberry preparation was made specifically for this investigation under highly standardized settings, with the primary ingredients being dried, sieved bilberries (59.63%) and concentrated bilberry juice (25.90%). Small metal trays containing 40 g of the preparation were used to package it. For a total of six weeks, patients were given a daily bilberry preparation dose of 160 g (4 trays per day), equal to 95 g dry weight (similar to about 600 g fresh fruit, assuming an 80–85 percent water content in fresh bilberries). Patients were instructed to avoid eating or drinking for one hour before and after ingesting bilberries. Endoscopic and histologic disease activity, as well as fecal calprotectin levels, were considerably reduced in the study participants after six weeks, suggesting that anthocyanins could be used as an alternative treatment human IBD patients. Both the feces and the tongue of all individuals had a dark bluish to black staining (one patient furthermore reported slight discoloration of the teeth). Mild dyspeptic symptoms were noted by one patient (heartburn). Furthermore, 33% of patients complained of mild to moderate flatulence. There were no major clinical adverse events or changes in the safety laboratory indicators that we noticed. Studies performed in healthy dogs confirmed the strong anti-oxidant activity of bilberry extract [87] but no data on the effects on dogs and cats affected by CE is available. 

#### 5.1.8. *Camellia sinensis*

*Camellia sinensis* (*C. sinensis*) is a species of evergreen shrubs or small trees. The leaves and leaf buds are used to produce tea (yellow tea, green tea, oolong tea, white tea, dark tea, and black tea). It has been demonstrated to have anti-inflammatory effects on lipopolysaccharide-stimulated macrophages and DSS-induced colitis in mice, reducing the oxidative stress. The researchers hypothesized that *C. sinensis* could be used as a safe and effective dietary strategy in preventing and treating human UC [41,88]. One study reports an anti-diarrheal effect of *C. sinensis* in children suffering from nonbacterial diarrhea [89,90]. Studies in dogs confirm the antioxidant properties of this plant [91,92] but no data on its effect on dogs and cats affected by CE is available in literature.

#### 5.1.9. *Citrus*

*Citrus* fruit and juices are among the most common phenolic rich dietary sources. For example, diosmetin is a natural flavonoid molecule found in citrus plants. It has a number of pharmacological properties, but little is known about its impact on CID. The therapeutic effects of diosmetin on mice models of chronic and acute colitis were investigated in a recent study [93]. They discovered that diosmetin treatment significantly reduced colon oxidative damage by regulating intracellular and mitochondrial reactive oxygen species levels. No information on the effects on dogs and cats affected by CE is available.

#### 5.1.10. Pomegranate

Pomegranates are not citrus fruits, despite their citrus flavor. They do not come from the same plant family and cannot be considered cousins. Their juice, however, can be blended to make a refreshing drink that is high in critical vitamins.

In mouse models of IBD, pomegranate fruit administration reduced colon tissue damage, antioxidant status, and inflammation [94]. In a DSS-induced colitis model, pomegranate extract was demonstrated to lessen the severity of colitis by modulating the gut microbiota and down-regulating COX-2, PTGES, iNOS, and PGE2 expression [94]. 

We found no in vivo studies on the use of citrus in the management of CE in humans, dogs, and cats.

### 5.2. Trace Elements

Two relevant trace elements, zinc, and selenium have been discussed in this review because of their antioxidant and anti-inflammatory effects. Overall, trace element status in CID in human and veterinary medicine, appears to be a neglected subject, and further clinical trials are needed to investigate this issue more thoroughly [32,95,96].

#### 5.2.1. Zinc

Zinc is an essential mineral that is naturally present in some foods (e.g., beans, nuts, seafood). Zinc influences oxidative stress, immune response, and inflammation [97]. It appears that zinc deficiency is widespread among human patients with IBD, with a prevalence of 15 to 45% [98]. Provided that IBD is a chronic inflammatory disease linked to immune system function, controlling and maintaining normal zinc levels in IBD human patients appears to be important. Reduced serum zinc concentrations have been shown in preclinical models, human translational studies, and animal model studies to exacerbate inflammation; this effect may be mediated by a variety of pathophysiological mechanisms, including the increased production of pro-inflammatory cells, modulation of the inflammatory cytokine response, aggravation of mucosa leakage, and disruption of the epithelial barrier [99]. Zinc is a cofactor for many enzymes and is involved in a variety of important processes, including the protection against free radicals and the control of innate immunity by regulatory cells [100]. Zinc deficiency has been shown to cause oxidative stress in a variety of cells and tissues, and zinc supplementation may help prevent oxidative harm [101]. Zinc, as a cofactor of the antioxidant enzyme SOD1, is integrated into the cellular antioxidant protection mechanism and protects cells from oxidative stress by increasing GPx biosynthesis, inducing metallothionein synthesis, and inhibiting NADPH oxidase [102], which is one of the most important sources of free radical activity [103]. The zinc-cytoprotective enzymes metallothioneins are up-regulated in response to an inflammatory stimulus as direct oxidant scavengers. These proteins belong to a group of cysteine-rich small proteins that play a role in reducing ROS development during oxidative stress [104]. Zinc also seems to be involved in the mucosal barrier function of the intestine. Zinc deficiency increases occludin proteolysis and decreases claudin-3 expression. Since these proteins are involved in the formation of tight junctions between intestinal epithelial cells, it appears that low zinc levels will weaken the intestinal mucosal barrier [105]. Due to zinc’s anti-inflammatory and antioxidant properties, as well as its protective role in the pathogenesis of IBD and the high prevalence of zinc deficiency among IBD human patients, providing adequate zinc levels is likely to be beneficial for more successful treatment of this chronic disease.

Vomiting, diarrhea, fever, lethargy, muscle discomfort, and stiffness are all symptoms of zinc toxicity, as are anemia, copper insufficiency, and kidney injury. In humans, the fatal dose of intravenous zinc is unknown. The National Institute of Health’s (USA) upper safe daily oral intake limit for elemental zinc is 40 mg/d, while the European Food Safety Authority’s lower limit is 25 mg/d. Zinc has been used as a parenteral nutrition component at levels ranging from 5 to 22 mg/d without any known negative effects [106]. 

In dogs and cats, zinc is known to be necessary for several normal functions linked to metabolic, enzymatic, and transcription factor activities, due to its antioxidant activity. Recommended zinc maximum level for complete dog food is 22.7 mg per 100 g dry matter (DM) [107]. Findings of a recent research demonstrated low levels of zinc in serum of dogs suffering from LPE (lymphocytic-plasmacytic enteritis), a kind of IBD. Thus, concentration of serum zinc could act as an indicator of LPE prognosis [108]. However, studies on zinc deficiency and the supplementation with this trace element in veterinary medicine are scarce [109,110]. Promising positive effects of the combination of zinc and other antioxidant elements on the oxidative stress caused by different diseases could be hypothesized. In one study, dogs fed with a diet supplemented with a preparation of selenium/zinc-enriched probiotics showed an increased total antioxidant capacity in the blood compared to the control group, supporting the antioxidant capacity of zinc [111]. Zinc has been used as a treatment in rats with experimentally induced liver cirrhosis and in copper toxicosis in dogs [112,113]. In another study, strong therapeutical potential of ZnO Nanoparticles has been reported to treat IBD in mice [114]. The synergic effect of zinc, silymarin, and vitamin E in a supplement tested in dogs has been shown to improve liver function [115]. In a group of dogs with IBD, the levels of zinc and magnesium were found to be lower compared to control dogs but no supplementation was tested. In addition, zinc was recommended for inflammatory and malabsorptive intestinal diseases in dogs where zinc absorption may be compromised [109]. Although preliminary studies on dogs have demonstrated lower zinc levels in various disorders i.e., skin, hepatic, renal, neurological, and behavioral disorders, therapeutic effects of zinc supplementation can be only hypothesized [116]. Unfortunately, the role of zinc deficiency and the usefulness of dietary supplementation with zinc in CE are yet to be investigated in dogs and cats. Minimum but not maximum zinc supplementation doses for dogs and cats have been established by the National Research Council (NRC, [117]) and safe doses for improving liver conditions have been tested in in vivo trials [115]. 

#### 5.2.2. Selenium

Selenium is a member of the sulfur family of elements and is naturally present in some foods (e.g., seafood, liver, and cereals). Selenium is primarily known for its antioxidant properties. The selenium-dependent enzyme glutathione peroxidase (GPX) is an essential antioxidant enzyme involved in the elimination of peroxides and hydroxyl free radicals produced during metabolism [118]. Selenium deficiency has been identified in human IBD patients in some studies [119]. Various selenoproteins are abundant in the intestine and may play a role in redox homeostasis control. The nuclear factor erythroid 2-related factor 2 (Nrf2)/Keap1 signaling pathway controls redox balance in cells. Nrf2 is a transcription factor that helps the body’s antioxidant defense system. The inflammatory processes in IBD appear to be weakened by Nrf2, and some studies have indicated that increasing its activity with compounds like polyphenols can help protect against intestinal inflammation [120]. LPS-induced oxidative stress and NO elevation can be decreased by selenium: NO is involved in the pathogenesis of IBD and has the ability to stimulate TNF-alfa development [95]. Furthermore, selenium may be able to minimize the severity of IBD inflammation by inhibiting prostaglandin E2 (PGE2) pro-inflammatory activity [95]. Selenium has been shown to protect against lipid peroxidation, and selenium supplements have also been shown to be helpful in this regard [121]. It is unclear if selenium supplements will help human patients with IBD stay in remission but, given the important roles of selenium and selenoproteins in reducing oxidative stress, inhibiting inflammatory signaling pathways, and increasing the population of anti-inflammatory M2 macrophages, it is conceivable that appropriate selenium levels could help keep them in remission. A study was carried out to see if a combination of selenium and vitamin E could protect rats against experimental colitis caused by acetic acid [122]. In plasma and colon samples, researchers measured the activities of prolidase (PRS), catalase (CAT), total antioxidant capacity (TAC), myeloperoxidase (MPO), oxidative stress index (OSI), total oxidant status (TOS), and total thiol (T-SH). Selenium and vitamin E treatment reduced MPO activity in the colon (*p* < 0.05). Selenium and vitamin E enhanced TAC and T-SH in the colon (*p* < 0.05). According to these findings, selenium and vitamin E may play a significant role in the prevention of oxidative damage caused by acetic acid-induced inflammation.

Selenium consumption levels are typically between 11 and 280 g per day. In the typical population, selenium levels in plasma and blood are around 100 g/L. In healthy adults, urine concentrations range from 10 to 85 g/L. It has been discovered that human doses of 10 mg/kg and higher are linked to an increased risk of death. The intake of gun bluing compounds, which often contain selenous acid as well as other potentially poisonous substances, is linked to the majority of human fatal instances. The administration of organic selenium in the form of selenocysteine or selenomethionine was not linked to any cases of acute toxicity [123].

For a long time, selenium was considered solely as a toxic element in animals. Recommended selenium maximum level for complete dog food is 56.80 µg per 100 g dry matter (DM) [107]. Cases of intoxication in horses, hogs, cattle, and chicken were reported for several years but not in dogs where the no-observed-adverse-effect level (NOAEL) of organic selenium was set in beagle dogs [124]. Probably, carnivores are able to maintain a high level of selenium compared to other species of animals, like most farm animals [96]. An interest in studying this element in the lab started to highlight its importance for health. At the moment, research on this trace element mainly focuses on humans and farm animals [125]. There is still a need for further studies on selenium targeting different species, including dogs, cats, and other carnivores. In a study, dogs were fed with a diet supplemented with a preparation of selenium/zinc-enriched probiotics, the biomass of the final product was 26.83 g/L, organic Se concentration was 173.35 µg/g, organic Zn concentration was 4.38 mg/g, *Candida utilis* biomass was 8.69 lg colony-forming units (CFU)/mL, and *Lactobacillus* biomass was 9.61 lg CFU/mL). The study showed total antioxidant capacity in the blood increased, supporting the antioxidant capacity of selenium [111]. A recent review reported interesting results on the effects of a diet supplemented with selenium in dogs with cancer, reproductive problems, renal disease, and parasitical diseases [96]. Unfortunately, no specific studies on the effect of selenium alone or in combination with other nutraceuticals on CE in dogs or cats are available. 

### 5.3. Vitamins

#### 5.3.1. Vitamin A

The most notable member of the group of carotenoids found in the human diet is b-carotene (b-CAR), a naturally occurring provitamin A that is a significant source of vitamin A for humans. Dogs and cats should get enough vitamin A from their normal diet. Vitamin A is essential for maintaining the integrity of the gastrointestinal tract’s epithelial cell lining as well as regulating immune activity. Reduced carotenoid levels have been linked to increased indicators of inflammation and oxidative stress [17,18]. Both these issues play a role in the pathophysiology of UC in humans. When compared to persons with normal mucosa, human patients with UC have significantly low serum (b-CAR) values. A study found that b-CAR had a protective effect in a DSS-induced UC animal model by reducing inflammation, oxidative stress, fibrosis, and DNA damage. It also reduced colonic mucosal damage and prevented occludin, a tight junction protein, from being reduced in the colons of mice with DSS colitis [126]. Vitamin A toxicity is uncommon in humans, however, it can occur as a result of increased vitamin A intake or after retinoid injection for therapeutic purposes. When the blood concentration of retinol in the plasma exceeds 2.09 µM, hypervitaminosis is diagnosed. The overuse of dietary supplements is frequently linked to toxicity. Chronic toxicity can develop with a long-term ingestion of 10 mg/day of vitamin A in adults and 7.5–15 mg/day in children for several months. Intakes of less than 30 mg/day (25,000–30,000 IU/day) are unlikely to cause toxicity [127] Vitamin A has functions in supporting vision, bone growth, reproduction, cellular differentiation, and immune response in dogs [128]. Dogs and especially cats have no capacity to synthesize vitamin A from b-CAR, as happens in humans. The required amounts of Vitamin A have been described in a study, given that Vitamin A is necessary for growth, maintenance, and lactation in dogs and cats [129]. Recommended Vitamin A maximum level for complete dog food is 40,000 IU (nutritional) per 100 g dry matter (DM) [107] No specific trials assessing the effect of Vitamin A in dogs and cats with CE were found in the literature.

#### 5.3.2. Vitamin E

Vitamin E is an essential vitamin, and a-tocopherol is the most active available form. Vitamin E is a lipophilic antioxidant that protects cellular membrane lipids from peroxidation, reduces the production of free radicals, and has anti-inflammatory properties. Enemas containing vitamin E reduced the disease symptoms of mild and moderately active human patients affected by UC in a clinical investigation [130]. Normally, cats and dogs experience oxidative damage that leads to chronic diseases. Vitamin E aids in preventing free radical damage. Findings of a study demonstrated that enhanced dietary levels of Vitamin E promote anti-oxidation and reduce oxidative damage in dogs and cats [131]. Dogs and cats should get enough vitamin E from their normal diet. Vitamin E can often be associated with Vitamin C in pets’ diet in order to increase antioxidant activity. No specific trials assessing the effect of Vitamin E in dogs and cats with CE were found in the literature [131].

#### 5.3.3. Vitamin C

Vitamin C is a group of related water-soluble substances (ascorbic acid, L-ascorbic acid, ascorbate, L-ascorbate). It is an antioxidant-rich natural substance that is utilized as a health supplement. The effects of intraperitoneal injection of high-dose vitamin C (4 g/kg) on DSS-induced UC were investigated in a study. High-dose vitamin C delivery reduced interleukin-6, hydrogen peroxide (H_2_O_2_), tumor necrosis factor-alfa, and iron levels in the blood. High-dose vitamin C delivery, on the other hand, raised the levels of H_2_O_2_ and iron in the colon, as well as the amount of terminal deoxynucleotidyl transferase-mediated deoxyuridine triphosphate nick-end labeling-positive cells. In mice given high doses of vitamin C, the expression of collagen type I, fibroblasts, and collagen type III increased. These findings imply that taking a high-dose vitamin C supplement can improve UC inflammation [132]. Vitamin C deficiency has also been linked to an increased risk of osteoporosis in human patients suffering from IBD [133].

Research suggests that both healthy dogs and cats are able to synthesize vitamin C independently from diet, so it is not recommended to add this vitamin in the diet [134]. Studies on dogs and cats have assessed vitamin C levels in patients with various medical disorders. The majority of these studies are focused on diseases associated with increased oxidative stress, including cancer, cardiovascular, renal, and infectious diseases. Evaluation of the level of Vitamin C in sick animals and studies on the effects of diets supplemented with vitamin C are limited in the veterinary literature [134]. As reported in a recent review, vitamin C was used in cases of severe burn injury in sheep, ischemic-reperfusion injury in a model of renal transplant in dogs, shock/traumatic injury in pigs [134]. The effect of vitamin C as an antioxidant has been confirmed in several studies suggesting a promising use in animals with severe oxidative damage. No specific trials assessing the effect of this vitamin in dogs and cats with CE have been found in the literature. 

**Table 1 animals-12-00812-t001:** Main human trials testing the antioxidant effects of nutraceuticals in CID patients.

Nutraceutical	Human Trial [Reference]	Proved Effectiveness in Human CID Patients
Phytocomplex		
*Curcuma longa*	Yes [62]	More effective than placebo in keeping human patients with quiescent UC in remission
*Aloe vera*	Yes [70]	Produces clinical response more frequently than placebo and reduction in histological disease activity
*Boswellia serrata*	Yes [73,74]	Maintains remission over a prolonged period in UC patients
*Triticum aestivum*	Yes [78]	Efficient treatment for active distal UC as a single or adjuvant treatment
*Plantago ovata*	Yes [82]	Its seeds have been shown to be as effective as mesalazine in preventing UC relapse
*Serpylli herba*	no	NA
*Vaccinium myrtillus*	Yes [86]	Endoscopic and histologic disease activity, as well as fecal calprotectin levels, were considerably reduced in UC
*Camellia sinensis*	no	NA
*Citrus*	no	NA
Trace elements		
Zinc	no	NA
Selenium	no	NA
Vitamins		
Vitamin A	no	NA
Vitamin E	Yes [130]	Enemas containing vitamin E reduced the disease symptoms of mild and moderately active UC
Vitamin C	no	NA

NA: no data available, UC: ulcerative colitis.

## 6. Concluding Remarks

Although an increasing number of new drugs are entering the market, especially to treat IBD in humans and serious CE in pets, the efficacy and the safety profiles of the current medications are far from optimal. Because CID is similar in pets (cats and dogs) and humans, the best of both experiences can benefit both worlds. Animals can represent a model for humans (they have a shorter lifespan and diseases have a faster course), while the human medicine field has a cutting-edge approach useful for the veterinary sector.

This review highlights the strong evidence for the antioxidant effect of all the selected nutraceuticals in both human and veterinary medicine. Unfortunately, in vivo studies where nutraceuticals are specifically used for treating CID are lacking in humans and absent in animals. However, given their unquestionable antioxidant and anti-inflammatory properties, most of these substances can be considered as a promising alternative for regular treatments of CID. In this perspective, the use of dietary interventions and complementary feeds could be beneficial for our patients in terms of efficacy and safety. New randomized controlled trials are needed to confirm their usefulness.

## Figures and Tables

**Figure 1 animals-12-00812-f001:**
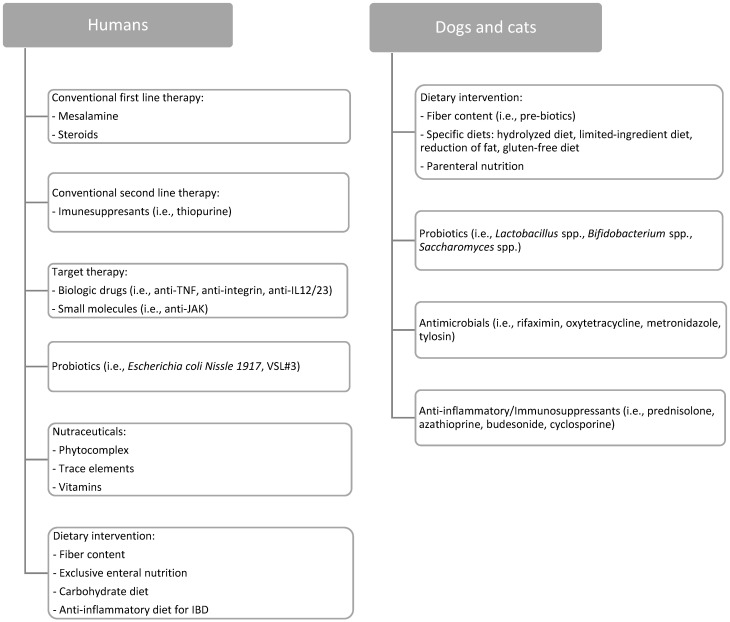
Stepwise approach to chronic intestinal disorders in humans and pets. UC = ulcerative colitis.

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
