# Peer review of "Chronic Intestinal Disorders in Humans and Pets: Current Management and the Potential of Nutraceutical Antioxidants as Alternatives"

_animals, 2022, doi:10.3390/ani12070812_

Round 1

Reviewer 1 Report

Table 1 is not well understood. It is not clear why the comparison is made with human medicine. 

Author Response

Dear Reviewer, 

Thank you for your comment.

Attached you can find our Answer.

Regards, 

Dr. Martello & co-authors

Reviewer 2 Report

Alongside superfoods, nutraceuticals are a hot topic in human and animal nutrition. Against this background, it is commendable that the authors try to give an overview.

First, a definition of chronic intestinal disorders is given and the background of this disease is examined. This makes sense and is necessary in order to understand the mode of action of nutraceuticals later on.

However, some of the terms were not clearly assigned.

On the one hand, it is formulated that IBD in pets is not an adequate term for what is happening, but that the term CE should be used instead. In the following text, however, the authors themselves use the term IBD.

In chapter 4 (Dietary Interventions) the authors are very superficial. With regard to fibres (chapter 4.1.1), a distinction must be made between easily digestible fibres and fibres that are difficult to digest. Fibres that are difficult to digest stimulate intestinal peristalsis and thus the expulsion of pathogenic germs. Easily digestible fibres (and here the authors should name examples such as pectins in carrots etc.) are degraded to volatile fatty acids, which serve as a nutrient substrate for the microbiota as well as the enterocytes.

The explanations of the nutraceuticals themselves also remain very superficial. Often neither the value-determining ingredient nor the mode of action is mentioned. In the case of some substances, it must also be taken into account that they have a toxic effect in higher doses. For this reason (as well as from an environmental point of view), feed legislation provides for maximum permitted levels for these additives. This applies, for example, to the components vitamin A, zinc and selenium.

The latter is also often used in combination with vitamin E to scavenge free radicals. This is not addressed here.

What is particularly serious, however, is the fact that no doses are given for any substance that were used in the individual studies and what effects were observed. A complete table with the data would have been desirable. Thus, the review has more the character of a textbook without actually being able to transfer data into practice.

In addition to these general impressions, I would like to address some specific comments below:

Line 22  alternative… for what

Line 33  none in animals… are you sure?

Line 36  add “nutraceuticals”

Line 45  and can severely…

Line 45  CD involve the entire

Line 50  ulcers) and clinically

Line 61  and 4)…. [9] as well.

Line 76  immunosuppressiva

Line 79  omega-3-fatty acids

Line 89  in a dysfunction

Line 94 mechanism,

Line 106 vitamins A

Line 106 plasma and blood = repetition

Line 123 delete the comma behind years

Line 180 in several studies… could be demonstrated

Line 215 cases diets did

Line 218               define the fiber high or low soluble

Line 293               no, not the digestibility is influenced, the proteon is broken down in parts below 10 dalton, therefore the orf´ganism do not recognized them as protein

Line 298 … not fed up to now

Line 302               define the maximum tolerated fat level

Line 488               still not well known? That is not correct

Author Response

Dear Reviewer, 

Thank you for your comments.

Attached you can find our Answers.

Regards, 

Dr. Martello & co-authors

Reviewer 3 Report

Manuscript animals-1603629, entitled “Chronic intestinal disorders: current management and nutraceutical antioxidants in humans and pets”

Recommendation:       The above paper is not suitable for publication in its present form.

General Comments:

  • My main concern is that the majority of the data presented refers to humans and not to pets. For example, in section 5, none of the presented medicinal plants - nutraceuticals, trace elements or vitamins has been already used for the treatment of chronic enteropathies in pets. I think that neither title nor journal is appropriate for this review article.
  • Table 1 is not correctly presented. Please clarify
  • L439-443: Pomegranate is not a plant of citrus family
  • L490-492, 538-540: The beneficial effects shown are a result of the sole addition of Zn/Se or of their combined supplementation?

Specific comments:

L18: “…intestine for a period of at least three…”

L19: “disorders” instead of “conditions”

L20: “…been increased in the recent decade, since oxidative stress plays a key role in…”

L21: “In this review, the antioxidant properties of several…”

L23: “…veterinary medicine are highlighted. Unfortunately…”

L27-29: “…colitis (UC). On the other hand, the use of the general term chronic enteropathies (CE) is preferred in veterinary medicine. Different…”

L30: “is” instead of “will be”

L31 and throughout the text: “properties” instead of “proprieties”

L31-32: “…humans and animals. There is strong evidence regarding the antioxidant properties of the nutraceuticals…”

L34: “Despite this fact, the majority of the nutraceuticals described in the present article could be considered…”

L44: “The peak of onset is at the age of around 20-40 years old, but they…”

L46: “…to the anus and is characterized…”

L48: “On the other hand” instead of “Conversely”

L50: “…clinically followed by bloody…”

L54-55: “…is preferred instead of IBD to identify…”

L56:  “…or chronic) and inflammation…”

L71: “varied” instead of “have a different”

L74: Please delete “for CE patients”

L81: “…disorders [9,12].”

L83: “…antioxidant properties that possess established or…”

L84: “…in humans and animals.”

L96: “damage” instead of “harm”

L100-101: Please delete (repetition; L92-93)

L103: “is observed” instead of “was seen”

L114: “…there is evidence indicating the role of…”

L126: “previous” instead of ‘past”

L126-127: “…is difficult to be established, since the pathogenesis of the disease is not easily understandable.”

L140: “symptoms” instead of “activity”

L141: “…function is unclear, however the most accepted theory is that it…”

L159: “previous” instead of “past”

L186: “percentage” instead of “fraction”

L213-216: “…of diet manipulation that in many cases resulting in a promising long term outcome (>6 months). In more severe cases, where a long-term positive effect is not observed, antibiotic or immunomodulants treatments should be added [5].”

L216: “reported” instead of “seen”

L217: “Despite the fact that most of…”

L218: “consisting of” instead of “having”

L222: “However” instead of “Further”

L223: Please delete “diet”

L225: Please delete “as shown below”

L234-235: “…a prebiotic function by increasing…”

L239: “In the existing scientific literature…”

L245: “suggest” instead of “observe”

L248: “…for many weeks, making adherence challenging and unpredictable. This…”

L258: “…are allowed in the SCD.”

L261: “not permitted” instead of “eliminated”

L262: “SCD” instead of “This diet”

L266-267: “…as a result of specific food restriction, especially dairy…”

L292: “…have been successfully used in…”

L310: Please rephrase

L313-314: “This type of nutrition keeps under control the daily intake of the different nutrients and contributes to bowel rest.”

L331: “…medicine (CAM) application has been found to be more common in…”

L333: “assist in the treatment of” instead of “help to treat”

L335-336: “…properties of the used plants. Several studies in dogs and cats reported the use of these natural ingredients as promising to manage diseases.”

L343: “Turmeric’s active…”

L345: “…it helps in the improvement of colitis.”

L353: “Meriva” ?

L357: “…succulent plant. The leaves…”

L358: “…that contains the most bioactive…”

L381: “…and anti-diabetic agent in dogs in…”

L402: “implemented” instead of “found”

L414: “that exert” instead of “showing”

L426: “…and DSS-induced colitis in mice, reducing…”

L433:  “rich” instead of “reach”

L453: Please delete (repetition; next sentence)

L469-470: “…NADPH oxidase [94] that is one of  the most important sources of free radical activity [95]. The zinc…”

L471-472: “…enzymes metallothioneins are up-regulated in response to an inflammatory stimulus as direct oxidant scavengers. These proteins…”

L477: “Due to” instead of “Given”

L482: “necessary” instead of “very relevant”

L483: “…factor activities, due to its antioxidant…”

L502: Please delete “only promising”

L503: “…be only hypothesized [106].”

L529: “…it is conceivable that appropriate selenium levels can help keeping them in remission.”

L532: “…where the no-observed-adverse-effect level (NOAEL) of organic…”

L566-567: “Vitamin E is an essential vitamin and the a-tocopherol is its most active available form. Vitamin E…”

L592-593: Please delete “has been made”

Author Response

(The authors gave the same response as above.)

Round 2

Reviewer 2 Report

The revision of the manuscript shows that the authors have worked intensively on the notes. On the one hand, the group of dietary fibers was broken down according to their dietary effect. In the case of the additives, the dosages used have been supplemented so that the reader now also receives valuable inforamtions.

Even though the revision of the manuscript certainly required a great deal of work (for which I would like to expressly thank the authors), the manuscript has been enhanced by the additional information and now represents a valuable source of information for practitioners.

Author Response

We would like to thank the Reviewer for the comments received that helped improving the Manuscript. 

Best wishes, 

Dr. Martello and co-authors

Reviewer 3 Report

Authors made all the necessary amendments. 

I think that before the acceptance of the article, the title should be somehow modified in order to reflect the content. I suggest "Chronic intestinal disorders in humans and pets: current management and the potential of nutraceutical antioxidants as alternatives"

Author Response

We would like to thank the Reviewer for the comments received that helped improving the Manuscript. 

We amended the title as suggested.

Best wishes, 

Dr. Martello and co-authors
